# Living Donor Kidney Transplantation Improves Graft and Recipient Survival in Patients with Multiple Kidney Transplants

**DOI:** 10.3390/jcm9072118

**Published:** 2020-07-05

**Authors:** Maria Irene Bellini, Aisling E Courtney, Jennifer A McCaughan

**Affiliations:** Regional Nephrology and Transplant Unit, Belfast City Hospital, 51 Lisburn Road, Belfast BT9 7AB, UK; aisling.courtney@belfasttrust.hscni.net (A.E.C.); jennifer.mccaughan@belfasttrust.hscni.net (J.A.M.)

**Keywords:** living donation, repeated kidney transplantation, graft survival, prolonged ischaemic time, patient survival, pre-emptive transplantation

## Abstract

Background: Failed kidney transplant recipients benefit from a new graft as the general incident dialysis population, although additional challenges in the management of these patients are often limiting the long-term outcomes. Previously failed grafts, a long history of comorbidities, side effects of long-term immunosuppression and previous surgical interventions are common characteristics in the repeated kidney transplantation population, leading to significant complex immunological and technical aspects and often compromising the short- and long-term results. Although recipients’ factors are acknowledged to represent one of the main determinants for graft and patient survival, there is increasing interest in expanding the donor’s pool safely, particularly for high-risk candidates. The role of living kidney donation in this peculiar context of repeated kidney transplantation has not been assessed thoroughly. The aim of the present study is to analyse the effects of a high-quality graft, such as the one retrieved from living kidney donors, in the repeated kidney transplant population context. Methods: Retrospective analysis of the outcomes of the repeated kidney transplant population at our institution from 1968 to 2019. Data were extracted from a prospectively maintained database and stratified according to the number of transplants: 1st, 2nd or 3rd+. The main outcomes were graft and patient survivals, recorded from time of transplant to graft failure (return to dialysis) and censored at patient death with a functioning graft. Duration of renal replacement therapy was expressed as cumulative time per month. A multivariate analysis considering death-censored graft survival, decade of transplantation, recipient age, donor age, living donor, transplant number, ischaemic time, time on renal replacement therapy prior to transplant and HLA mismatch at HLA-A, -B and -DR was conducted. In the multivariate analysis of recipient survival, diabetic nephropathy as primary renal disease was also included. Results: A total of 2395 kidney transplant recipients were analysed: 2062 (83.8%) with the 1st kidney transplant, 279 (11.3%) with the 2nd graft, 46 (2.2%) with the 3rd+. Mean age of 1st kidney transplant recipients was 43.6 ± 16.3 years, versus 39.9 ± 14.4 for 2nd and 41.4 ± 11.5 for 3rd+ (*p* < 0.001). Aside from being younger, repeated kidney transplant patients were also more often males (*p* = 0.006), with a longer time spent on renal replacement therapy (*p* < 0.0001) and a higher degree of sensitisation, expressed as calculated reaction frequency (*p* < 0.001). There was also an association between multiple kidney transplants and better HLA match at transplantation (*p* < 0.0001). A difference in death-censored graft survival by number of transplants was seen, with a median graft survival of 328 months for recipients of the 1st transplant, 209 months for the 2nd and 150 months for the 3rd+ (*p* = 0.038). The same difference was seen in deceased donor kidneys (*p* = 0.048), but not in grafts from living donors (*p* = 0.2). Patient survival was comparable between the three groups (*p* = 0.59). Conclusions: In the attempt to expand the organ donor pool, particular attention should be reserved to high complex recipients, such as the repeated kidney transplant population. In this peculiar context, the quality of the donor has been shown to represent a main determinant for graft survival—in fact, kidney retrieved from living donors provide comparable outcomes to those from single-graft recipients.

## 1. Introduction

The proportion of end-stage renal disease (ESRD) patients with a failed kidney transplant is increasing each year [1,2]; 25% of patients on the US kidney transplant waiting list have a failed transplant [3] and in the Eurotransplant area, the number of patients being re-waitlisted after returning to dialysis steadily ranges between 17.9% and 18.9% [4].

Morbidity and mortality for patients with a failed kidney transplant on dialysis is high; there is a two- to three-fold risk of death compared to patients with a functioning graft and a median recipient survival after graft failure of three years [3,5]. This increased mortality [6] relates to higher rates of cardiovascular, neoplastic and infective events, in which the burden of immunosuppressive therapy is no longer counterbalanced by the benefits of a working kidney transplant [7]. The single modifiable factor which has the greatest impact on recipient survival in this group is the time to re-transplantation [6]; however, the suboptimal outcome of repeated kidney re-transplantation has generated increasing debate regarding the overall management and resource allocation within this subgroup. Nevertheless, previous reports have shown that there is a significative survival benefit after repeat deceased donor kidney transplantation over remaining on the waiting list, due to significative improvement in better immunological screening, crossmatching, HLA matching, post-operative management and immunosuppression protocols [8], although the overall outcome remains impaired by an inferior survival compared to first kidney transplant recipients [9].

Yet, other fundamental outcome drivers, for example the impact of a high-quality living donor graft, have not been fully investigated in the peculiar context of repeated kidney transplantation.

One of the primary advantages in fact of receiving a kidney from a living donor is that the organ is generally healthier and more resistant to the occurrence and extension of the subsequent ischaemic reperfusion injury. To become eligible, living donors undergo full screening of their kidney function, tissue and immunological compatibility with the recipient and a comprehensive overall physical health check. This is in contrast with grafts retrieved from deceased donors, where already the stress and the damage related to the death of the individual determine a systemic storm summing up to the usual longer time in cold storage to allow retrieval and transfer between different teams and hospitals. All together, these factors can temporarily reduce and potentially compromise the organ function irreversibly [10].

The incidence of delayed graft function in kidneys from deceased donors varies, but is overall as high as 30%; it might also take weeks before the recipient is fully dialysis independent [11,12], thus the recipients are more exposed and vulnerable in the post-operative period. Conversely, kidneys from living donors tend to function immediately, reducing the risk of hospitalisation and renal replacement therapy after transplant to less than 4% [13] and setting in this way the recipient for the best short- and long-term outcome.

Why is it so important an immediate graft function? As stated above, the more complex the procedure, the higher the likelihood of the prolonged ischaemic insult and the resultant impact on challenging recipients. Previously failed grafts, a long history of comorbidities, side effects of long-term immunosuppression and previous surgical interventions are common characteristics in the repeated kidney transplant population, leading to significant immunological and technical challenges.

Intuitively, it might initially seem sensible to withdraw immunosuppression in patients after graft failure to reduce the risk of cardiovascular, neoplastic and infective complications. However, for those who are fit for a subsequent transplant, this commonly results in a high degree of sensitisation to HLA, namely the production of donor-specific HLA antibodies (DSA) and other panel-reactive antibodies (PRAs) [14]. This reduces access to compatible donors and may result in an extremely prolonged wait for transplantation, with the associated morbidity and mortality [15]. This increased waiting time might also have a worse synergistic effect with the often extended ischemic time, due to the technical challenges associated with re-transplantation, namely adhesions from previous surgery, difficulties accessing the iliac vasculature or earlier manipulation of the bladder to establish the ureterovesical anastomosis [16].

The aim of this study is to analyse the experience of repeated kidney transplantation in our institution over 50 years, with a focus on outcomes in recipients of first, second and third/fourth kidney transplants and factors which impact these outcomes.

## 2. Methods

All recipients of kidney transplants in Northern Ireland between 1968 and 2019 were included in the analysis. Recipients were identified using a prospectively maintained database which records data on all transplant recipients. Recipients were followed up until death or 1 September 2019. The clinical and research activities being reported are consistent with the principles of the Declaration of Helsinki and comply with the Declaration of Istanbul. Approval for this study was granted by the Regional Ethics Committee (12/NI/1078).

Death-censored graft survival was measured from time of transplant to graft failure (return to dialysis) and censored when a patient died with a functioning graft. Duration of renal replacement therapy was expressed as cumulative time per month. Recipient survival was measured from transplantation to death. Pre-transplant sensitisation levels were expressed as a calculated reaction frequency (cRF), which is calculated as the proportion of the last 10,000 UK, blood group-identical, deceased donors to which the patient has DSAs. Recipients were considered highly sensitized if they had a cRF greater than 85%.

Immunosuppression: No routine induction was used in any era. Maintenance regimen was on prednisolone and azathioprine before 1989; from 1989 to 1998, cyclosporine was introduced and patients commenced on triple therapy; in 1998, mycophenolate mofetil replaced azathioprine in the triple-therapy regimen; and from 2000, tacrolimus replaced cyclosporine. Overall, the majority of patients were maintained on two agents in the long term, with 25% on the calcineurin inhibitor-free regimen.

Living donors: Numbers of living donors performed varied according to the decade considered: 3.3% from 1968 to 1977; 9.8% from 1978 to 1987; 4% from 1988 to 1997; 10.6% from 1998 to 2007 and 55.3% from 2008 to 2017.

### Statistical Analysis

Continuous variables are presented as the mean ± standard deviation. Analysis of variance and *t* test were used to compare continuous variables between groups. For nominal or non-parametric variables, the Pearson χ^2^ test was performed. Kaplan–Meier and Cox regression analyses were applied for survival analysis. In a multivariate analysis for death-censored graft survival, factors previously associated in our population were included: decade of transplantation, recipient age, donor age, living donor, transplant number, ischaemic time, time on renal replacement therapy prior to transplant and HLA mismatch at HLA-A, -B and -DR. In the multivariate analysis of recipient survival, diabetic nephropathy as primary renal disease was also included. Confidence interval was set to 95%, and *p* was considered significant at less than 0.05. Analysis was performed using SPSS (IBM SPSS Statistics for Windows, Version 20.0; IBM Corp, Armonk, NY, USA).

## 3. Results

A total of 2395 kidney transplant recipients were included: 2062 (83.8%) received a 1st kidney transplant, 279 (11.3%) received a 2nd kidney transplant, 46 (1.9%) received a 3rd kidney transplant and 8 (0.3%) received a 4th kidney transplant. The outcomes of the 3rd and 4th kidney transplants were grouped together (3rd+).

Table 1 summarises donor and recipient characteristics. Recipients of 3rd+ kidney transplants were significantly more likely to receive a living donor kidney (*p* < 0.0001).

In total, 99% of recipients were White. Recipients of repeated kidney transplants were more likely to be male (*p* = 0.006) and were younger (*p* < 0.001): mean age of 1st KTRs was 43.6 ± 16.3 years, versus 39.9 ± 14.4 for 2nd and 41.4 ± 11.5 for 3rd+ KTRs. Furthermore, these patients were also significantly more sensitised, with an increasing cRF from 15% (1st transplant) to 54% (2nd transplant), to 76% (3rd+ transplant) (*p* < 0.0001). As a consequence, there was also an association between multiple kidney transplants and better HLA match at transplantation (*p* < 0.0001). The pre-emptive rate was significantly lower in recipients of multiple transplants (*p* < 0.0001).

### 3.1. Surgical Information

All kidney transplants were performed extraperitoneally and graft nephrectomy was only performed in four cases: one in relation to uncontrolled antibody mediated rejection with systemic involvement, one following a catastrophic bleed, one simultaneously to the implant and one to create space for a potential 4th graft. The final patient had had multiple gynaecology procedures and the peritoneal content would not have been easily mobilised to allow graft implantation.

Only one major bleeding event occurred that required graft nephrectomy (2nd implant), but the recipient underwent successful implantation of a 3rd graft three years later. Urological complications were not recorded.

### 3.2. Death-Censored Graft and Recipient Survival

Figure 1 shows death-censored graft survival, with a median of 328 months for 1st kidney transplant recipients (KTRs) in blue, 209 months for 2nd KTRs in green and 150 months for 3rd+ KTRs in red (*p* = 0.04).

Death-censored graft survival remained significantly different between the three groups in the case of deceased donor transplants (Figure 2a), but there was no significant difference in death-censored graft survival between the groups in living donor transplantation (Figure 2b).

Recipient survival was comparable between the three groups (*p* = 0.59), with a median of 234 months for 1st KTRs in blue, 256 months for 2nd KTRs in green and 298 months for 3rd+ KTRs in red (Figure 3). The 10 year recipient survival for all groups exceeded 70%.

In multivariate analysis, earlier decade of transplantation, older donor and recipient age, longer ischemic time, and transplant number were significantly associated with death-censored graft loss. Living donor transplantation was associated with improved death-censored graft survival (Table 2).

For recipients of 3rd+ transplants, the association with a living donor is the only factor associated with death-censored graft survival (Table 3).

Despite small numbers, a living donor transplant was associated with a 90% reduction in death-censored graft loss.

In multivariate analysis for recipient survival, significant factors were decade of transplant, recipient age, recipient primary disease of diabetic nephropathy, duration of RRT pre-transplant, living donor, donor age and ischaemic time (Table 4).

## 4. Discussion

This study investigated the outcomes of repeated kidney transplantation at our institution and demonstrated excellent graft and recipient outcomes (Figure 1 and Figure 3), despite a significantly more sensitised population and a longer vintage of ESRD. There was also an association between multiple kidney transplants and better HLA match at transplantation (*p* < 0.0001); this is unsurprising, as more highly sensitised patients require better matched kidneys. Our results are in contrast to a recent European multicentre analysis reporting that mortality and graft loss after 3rd+ KTRs were higher as compared to 1st KTRs, despite receiving grafts with more favourable HLA matches [17]. More in detail, Assfalg et al. analysed the outcomes of 1464 patients from 42 centres in the Eurotransplant area who received a third or fourth kidney transplant in the period 1996–2010, confirming a younger age compared to first transplant recipients, a more frequently favourable HLA match, but a higher rate of graft loss, death with functioning graft and primary non-function. Their conclusion was, therefore, to set an upper limit for the number of sequential transplantations in order to consider also the prospect of success of transplantation. In our study, it was confirmed that there is a significant difference in death-censored graft survival by number of transplants, as shown in Figure 1, with a median graft survival of 328 months for 1st KTRs, 209 months for 2nd KTRs and 150 months for 3rd+ KTRs, but the death-censored graft survival remained significantly different in the case of deceased donor transplants (Figure 2a), but not after living donor transplantation (Figure 2b). This suggests that living donor transplantation confers a significant benefit in the context of repeated kidney transplantation and challenges the assumption that repeated transplant recipients as well as any other special group of patients should be a priori denied access to transplantation [18]. Important modifiable factors, such as the quality of the implanted graft or the time at which the operation is performed, could significantly affect the overall outcome and this should be taken into consideration when planning such a complex procedure and before fixing an arbitrary cut off number to waitlist transplant candidates.

In this regard, an extensive patient work up with multidisciplinary input is highly recommended to maximise the chances of successful waitlisting for the candidate, to be followed by a successful repeated kidney transplantation. Further, in our study, in fact, 3rd+ KTRs are overall better matched compared to 1st or 2nd KTRs (*p* < 0.001), reflecting the broad immunological work up. Yet, despite a similar finding to the study from Dabare et al. [19], where patient survival did not differ significantly by transplant number even considering third or fourth KTRs and, therefore, confirming the survival benefit for this population over remaining on dialysis, we disagree with the authors’ conclusion. In our present analysis, we observed a significant decline in graft survival only in the case of deceased donor grafts, with a progressive worsening survival curve in parallel with the progressive increase in repeated kidney transplant number (Figure 2a). The conclusion that regardless of the donor type, there is an inferior graft survival for the repeated kidney transplant population is not confirmed instead for grafts retrieved from living donors (Figure 2b). Therefore, the authors’ suggestion to use HLA-incompatible living donors and extended criteria deceased donors in the peculiar context of the repeated kidney transplant population is not justifiable from our experience.

Prolonged ischaemic time is significantly detrimental to long-term survival for deceased donor grafts [20], with preservation strategies being key for suboptimal and extended criteria deceased donor organs [12]. Often 3rd and 4th kidney transplants are associated with prolonged ischaemic times due to increasing technical complexity: KTRs of 3rd and 4th transplants may have difficult vasculature that often requires additional surgical time. The deceased donor kidney performs less well in this context, while better-quality living donor kidneys can tolerate the insult. In our centre, where there is a high rate of living donor transplantation, there is for this reason an even higher proportion of organs retrieved from living donors in the case of 3rd+ KTRs (Table 1). To overcome living donor shortage, broad educational campaigns aiming to educate and inform the general population [21] and particularly via social media, have demonstrated an increase in donation rates [22]. With the current organ donor shortage and more patients dying on the waiting list, living donor kidney donation seems, therefore, to satisfy and significantly contribute to expand the donor pool for the general population, and more specifically for those who might not survive a long waiting list time or a major operation, like in the case of the repeated kidney transplantation subgroup. In addition, every living donor transplant that occurs removes one person from the transplant waiting list, shortening the waiting list for a deceased donor transplant, too.

With increasing evidence of how the preservation time and modality significantly impact organs retrieved from marginal donors [10] and with an even increasing debate in how to safely implement the donor pool, without compromising recipient outcomes, until a general consensus on how to best treat and preserve deceased donor grafts [23], an effort to find alternative ways of influencing patient and graft survivals should be canvassed in the ethical attempt to provide the best renal replacement therapy to those who need it.

Another important advantage offered by living donation is that an elective operation allows diligent planning and the presence of additional surgical expertise for complex cases [24]. This might also contribute to better outcomes, independently from the quality of the donor [19], with a standardised elective procedure taking place at an optimal time and that, despite being a major operation, has a minor impact [25] on the recovery of the healthy donor, who usually can plan ahead for time off work, for family care for and for a full recovery.

With deceased donor transplantation, the surgery often takes place out of hours; additionally, emergency procedures usually carry out extra unanticipated risks [26], along with the impossibility to schedule the time and avoid the waitlist consequences and deterioration on the general health status of the candidate, who might even be transplanted after several years, because of the complex immunological status. Inevitably, the elderly and sickest candidates might, therefore, be more susceptible to the vicious cycle of repeated kidney transplantation, becoming not transplantable, with a significant drop out from the list, or a detrimental transplant outcome. In our opinion, this is, therefore, why living donor kidney donation is so fundamental to expand the donor pool: removing successfully a difficult transplant candidate from the transplant waiting list and ensuring that the next person on the list will not have to wait as long for a deceased donor transplant.

The preferred surgical approach in the case of repeated kidney transplantation at our centre is extraperitoneal to avoid ileus and expedite an enhanced recovery [27]. In the case of native polycystic kidney disease, further space for transplantation might become a challenge, and therefore the affected patients are more likely to undergo native nephrectomy before the planned transplant, as they are already vintage patients [28]. In our series, 13 patients (23.6% of the total population) underwent bilateral native nephrectomy.

Differently from native nephrectomy, kidney transplantectomy was rarely performed in the case of a failed graft. The British Transplant Society guidelines [29] suggest limited indications for graft nephrectomy: localised symptoms that are resistant to medical therapy, to create space for re-transplantation, to enable complete withdrawal of immunosuppression and where there is risk of graft rupture or graft malignancy. This caution with regard to graft nephrectomy is partly due to its immunological effect [30], as nephrectomy and the cessation of immunosuppression can precipitate the development of HLA antibodies (DSAs and PRAs), which limits access to re-transplantation. In our series, only four graft nephrectomies were recorded: because of antibody mediated rejection with systemic involvement, because of a bleeding catastrophe after the transplant and to create space for a potential further kidney transplant. The limited number of graft nephrectomy at our institution is in contrast with other centres’ experiences, estimated at approximately 40% from a Turkish report [31] and up to 75% in a UK single-centre experience from 2009 [32]. We tend to avoid, as a general principle, an additional operation, unless not strictly required, in consideration of the controversy affecting the immunological recipient status and the likelihood of finding a suitable match, with antibody absorption from the graft itself. [30,32]. As previously stated, in fact the graft nephrectomy would imply the cessation of the immunosuppression, giving rise to antibody production due to the persistence of donor antigen-presenting cells after the transplantectomy. Furthermore, with the evidence that HLA matching plays a fundamental role in the context of repeated kidney transplantation, from the present study and another large registry analysis [17], we think that a synergistic approach to optimise the recipient condition and general immunological status would better satisfy increased complexity at the moment of transplantation in the eventuality that a prolonged surgical time would be required to find a suitable implantation site. Once again, we emphasize the importance of a high-quality graft, like the one retrieved from a living donor, to better resist a prolonged ischaemic insult.

Finally, given the increased morbidity in patients with failed kidney transplants, special attention should be paid to the attainment of cardiovascular and other infective or malignant events [33], the main cause of death in the long term and also in particular for the repeated KTRs. In our centre, we acknowledge this extra care and, notably, recipient survival did not differ between the groups.

## 5. Limitations

The main limitation of this study is the retrospective data assessment from a single institution, subjected to selection bias by the nature of the study itself, with a change in the immunosuppression and overall management of kidney transplant recipients over time. There is also an immortal bias due to younger age and higher fitness of the recipients of repeated kidney transplantation, possibly leading to comparable patient survival between the subgroups. Nevertheless, the comprehensive assessment from the same institution over a period of more than 50 years shed light on the context of the failing kidney transplant population, for which the best treatment and donor selection remains unclear in terms of long-term outcomes.

## 6. Conclusions

In conclusion, a living donor kidney has the greatest impact on improving graft and recipient survival in patients with multiple kidney transplants. We recommend early work up of recipients with failing grafts to achieve pre-emptive transplantation and minimise time on dialysis, and early pursuit of a living donor option for these individuals.

In our view, it is fundamental to consider that not only recipients’ factors but also donors’ characteristics are strongly related to short- and long-term results after kidney transplantation and that the higher the risk represented by the recipient, as in the case of the repeated kidney transplant population, the more likely stress and damage in the immediate and longer follow up will occur, therefore potentially irreversibly compromising the graft.

Living donor kidneys represent an undervalued resource significantly impacting the transplant outcome for higher-risk candidates, where a standard donor is instead more likely to be affected by the recipient status.

## Figures and Tables

**Figure 1 jcm-09-02118-f001:**
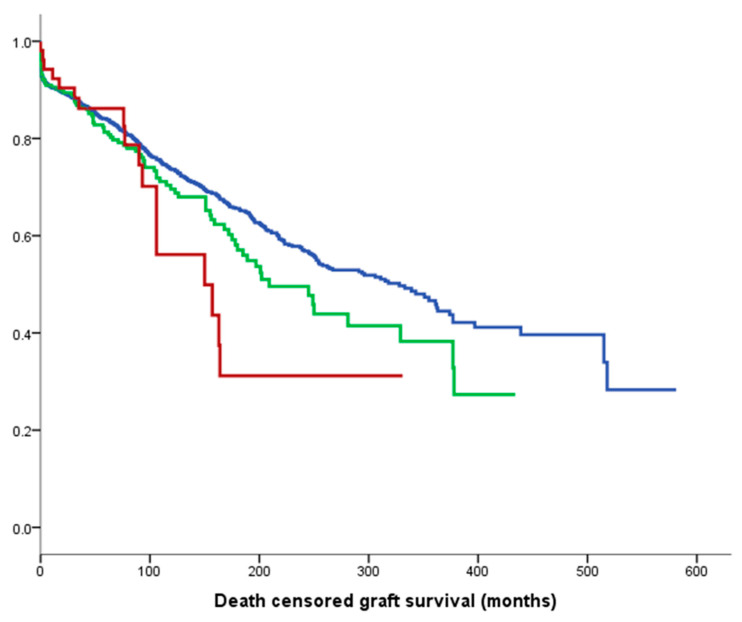
Median death-censored graft survival: 328 months for 1st graft (blue line), 209 months for 2nd (green line) and 150 months for 3rd+ (red line). (*p* = 0.04).

**Figure 2 jcm-09-02118-f002:**
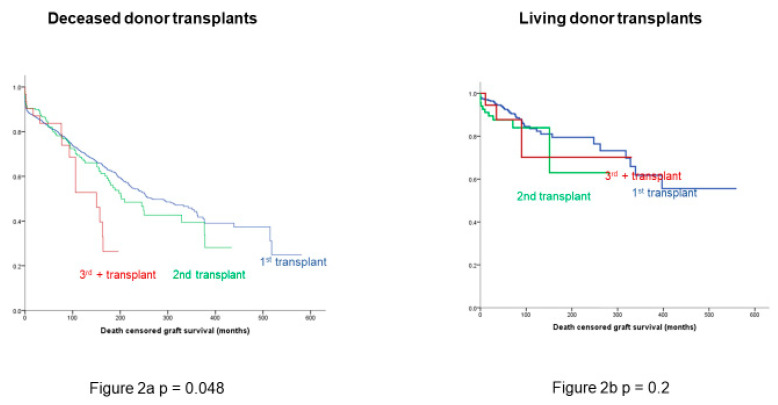
Difference in death-censored graft survival is seen in deceased donor transplants (**a**) but not in kidneys retrieved from living donors (**b**). Prolonged ischaemia is significantly detrimental to long-term survival in deceased donor grafts and 3rd and 4th transplants are associated with prolonged ischaemic times. These are marginal patients with difficult vasculature—marginal kidneys do less well in this context while good kidneys cope fine. Blue line: recipients of 1st kidney transplant; green line: recipients of 2nd kidney transplant; red line: recipients of 3rd kidney transplant.

**Figure 3 jcm-09-02118-f003:**
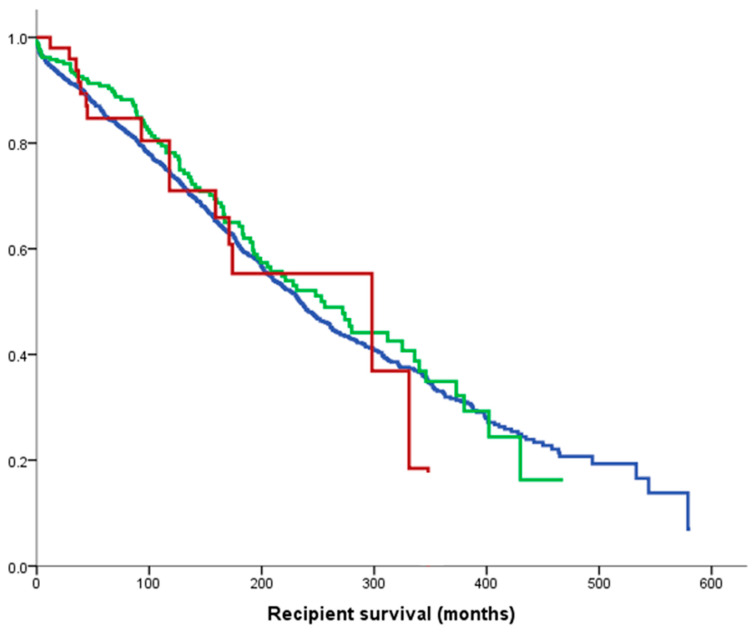
Median recipient survivals: 234 months for 1st graft (blue line); 256 months for 2nd (green line); 298 months for 3rd+ (red line) (*p* = 0.59).

**Table 1 jcm-09-02118-t001:** Demographics of kidney transplants performed in Northern Ireland in the period 1968–2019.

	1st Transplant *n* = 2062	2nd Transplant *n* = 279	3rd+ Transplant *n* = 54	*p* Value
**Recipient**				
Recipient age (mean+SD, years)	43.6 ± 16.3	39.9 ± 14.4	41.4 ± 11.5	0.001
Recipient sex (male)	1244 (60%)	187 (67%)	38 (73%)	0.006
Renal replacement therapy duration (mean+SD, months)	22.5 ± 26.6	124.7 ± 95.8	207.3 ± 106.8	<0.0001
Calculated reaction frequency (cRF) (mean+SD, %)	15.3 ± 29.7	54.1 ± 40.3	75.7 ± 34.5	<0.0001
**Donor**				
Donor age (mean+SD, years)	49.7 ± 9	39.4 ± 16.1	43.5 ± 16	0.166
Donor sex (male)	1159 (56%)	152 (54%)	28 (54%)	0.171
Living donor	549 (26%)	82 (29%)	22 (42%)	<0.0001
**Transplant**				
Pre-emptive transplantation	308 (14.9%)	22 (7.9%)	3 (5.6%)	<0.0001
HLA-A, -B, -DR mismatch (mean+SD, number)	2.4 ± 1.3	2 ± 1.4	1.4 ± 1.4	<0.0001
Ischaemic time (mean+SD, hours)	17.5 + 18	16.6 + 16.5	13 + 9.9	0.197

Donor age, donor sex and cold ischaemic time did not statistically differ between the groups.

**Table 2 jcm-09-02118-t002:** Factors associated with death-censored graft survival in all recipients on a multivariate analysis.

Covariate	HR	95% CI	*p* Value
Decade of transplant	0.84	0.78–0.90	<0.001
Recipient age (per decade)	0.90	0.86–0.96	<0.001
Living donor	0.61	0.45–0.83	0.002
Donor age (per decade)	1.14	1.08–1.20	<0.001
Transplant number	1.23	1.03–1.48	0.02
Ischaemic time (per 6h)	1.09	1.01–1.17	0.02

Time on RRT and HLA mismatch at HLA-A, - B and -DR were not significant and dropped out of model.

**Table 3 jcm-09-02118-t003:** Multivariate analysis for death-censored graft survival in 3rd+ recipients.

Covariate	HR	95% CI	*p* Value
Decade of transplant	0.98	0.49–1.96	0.9
Recipient age (per decade)	0.85	0.51–1.39	0.5
Living donor	0.10	0.01–0.89	0.04
Donor age (per decade)	0.76	0.63–1.40	0.8
Ischaemic time (per 6h)	0.57	0.30–1.10	0.08

**Table 4 jcm-09-02118-t004:** Multivariate analysis for recipient survival in all transplants.

Covariate	HR	95% CI	*p* Value
Decade of transplant	0.60	0.56–0.65	<0.001
Recipient age (per decade)	1.7	1.6–1.8	<0.001
Recipient primary disease diabetic nephropathy	2.8	2.3–3.5	<0.001
RRT pre-transplant (per month)	1.002	1.00–1.004	0.02
Living donor	0.75	0.56–1.00	0.05
Donor age (per decade)	1.07	1.02–1.1	0.03
Ischaemic time (per 6h)	1.1	1.04–1.2	0.002

HLA mismatch at HLA-A, -B and -DR and transplant number were not significant and dropped out of model.

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
