# Peer review of "Living Donor Kidney Transplantation Improves Graft and Recipient Survival in Patients with Multiple Kidney Transplants"

_jcm, 2020, doi:10.3390/jcm9072118_

Round 1

Reviewer 1 Report

Thank you authors for allowing me to review your prestigious work. The manuscript is well written and describes about the clinical impact of LDKTin patients with multiple KT. 

Introduction

  • No revisions needed.

Methods

  • Please describe the details for the multivariate analysis e.g. logistic regression vs. Cox regression? And, how which variables the results were adjusted for? An additional footnote for Table 2 stating which variables the multivariate analysis was adjusted for would be appreciated.

Results

  • Although Table 1 depicts the demographics of included patients, some variables are missing, such as sex, ethnicity, and co-morbidities. If authors could not provided those information in Table 1, authors can upload as supplemental material.
  • It seems like Table 2, 4 represent multivariate analysis of "all transplant". This can create a significant limitation when authors tried to draw a conclusion that "living donors improved graft/patient survival" because most patients were 1st transplant. Moreover, the content of using a living donor has been shown to have more benefits in every aspects of transplant outcomes. Authors do not need to prove such statement as it is redundant. In fact, the effect of living donors in "re-kidney transplant" is not so well described. That is why authors conducted this study, correct? I would recommend regrouping the patient as "1st tranplant" vs. "2nd transplant and more" and analyze with regression again to determine if living donors have clinical benefits. Authors do not need to delete the results for "all transplants". In contrast, authors can create another section reporting the results of "1st transplant" vs. "2nd transplant and more". This will provide a significant value to your work.

Discussion

  • Overall, the Discussion is well written. However, some improvements are warranted.
  • If authors decide to add the results as recommended above, the Discussion has to be updated accordingly.
  • The authors did not mention one thing; why living donor kidneys are better than deceased donors. The pathophysiology should be outlined as well.
  • There are more limitations. Retrospective observation design is subjected to selection bias by nature. Although the study included up to 50 years, this could also be the limitation. The management of KT patients had changed significantly as authors mentioned in the Methods. Thus, the observed effects may be underpowered.

Abstract

  • Please update accordingly once the manuscript is revised. 

Author Response

Thank you authors for allowing me to review your prestigious work. The manuscript is well written and describes about the clinical impact of LDKTin patients with multiple KT. 

We thank the reviewer for the time to review our manuscript and the valuable comments that significantly improve the quality of the work.

Introduction

  • No revisions needed.

Thank you

Methods

  • Please describe the details for the multivariate analysis e.g. logistic regression vs. Cox regression? And, how which variables the results were adjusted for? An additional footnote for Table 2 stating which variables the multivariate analysis was adjusted for would be appreciated.

Thank you. Details of the multivariate analysis have been added: In multivariate analysis for death censored graft survival, factors previously associated in our population were included: decade of transplantation, recipient age, donor age, living donor, transplant number, ischaemic time, time on renal replacement therapy prior to transplant and HLA mismatch at HLA-A, B and DR. In the multivariate analysis of recipient survival, diabetic nephropathy as primary renal disease was also included. The variables adjusted for were included in each table.

Results

  • Although Table 1 depicts the demographics of included patients, some variables are missing, such as sex, ethnicity, and co-morbidities. If authors could not provided those information in Table 1, authors can upload as supplemental material.

Thank you. Table 1 includes information on sex. 99% of recipients were White; this has now been added.

  • It seems like Table 2, 4 represent multivariate analysis of "all transplant". This can create a significant limitation when authors tried to draw a conclusion that "living donors improved graft/patient survival" because most patients were 1st transplant. Moreover, the content of using a living donor has been shown to have more benefits in every aspects of transplant outcomes. Authors do not need to prove such statement as it is redundant. In fact, the effect of living donors in "re-kidney transplant" is not so well described. That is why authors conducted this study, correct? I would recommend regrouping the patient as "1st tranplant" vs. "2nd transplant and more" and analyze with regression again to determine if living donors have clinical benefits. Authors do not need to delete the results for "all transplants". In contrast, authors can create another section reporting the results of "1st transplant" vs. "2nd transplant and more". This will provide a significant value to your work.

Thank you for your valuable comment. We believe that challenges sum up with an increasing number of kidney transplants, therefore we prefer to emphasize how these are taken into account analysing the single retransplant groups and show that the number of retransplant is not the main determinant in the long-term outcome, but that instead a high quality donor plays a major role in this challenging context.

Discussion

  • Overall, the Discussion is well written. However, some improvements are warranted.

Thank you

  • If authors decide to add the results as recommended above, the Discussion has to be updated accordingly.

Thank you, we have extended the discussion

  • The authors did not mention one thing; why living donor kidneys are better than deceased donors. The pathophysiology should be outlined as well.

Thank you, we have added the following in the introduction:

Yet, other fundamental outcome drivers, as for example the impact of a high-quality living donor graft, have not been fully investigated in this peculiar context of repeated kidney transplantation.

One of the primary advantages in fact of receiving a kidney from a living donor is that the organ is generally healthier and more resistant to the occurrence and extension of the subsequent ischaemic reperfusion injury. To become eligible, living donors undergo full screening of their kidney function, tissue and immunological compatibility with the recipient and a comprehensive overall physical health check. This is in contrast with grafts retrieved from deceased donors, where already the stress of the death of the individual, with the consequent cytokine storm in addition to the usual longer time in cold storage to allow retrieval and transfer between different teams and hospitals, can temporarily reduce and potentially compromise the organ function.

The incidence of delayed graft function in kidneys from deceased donors varies, but is overall as high as 30%; it might also take weeks before the recipient is fully dialysis-independent, thus the recipients are more exposed and vulnerable in the post-operative period. Conversely, kidneys from living donors tend to function immediately, reducing the risk of hospitalisation and renal replacement therapy after transplant to less than 4%  and setting in this way the recipient for the best short and long term outcome.

  • There are more limitations. Retrospective observation design is subjected to selection bias by nature. Although the study included up to 50 years, this could also be the limitation. The management of KT patients had changed significantly as authors mentioned in the Methods. Thus, the observed effects may be underpowered.

Thank you we have added the following: The main limitation of this study is the retrospective data assessment from a single institution, subjected to selection bias by the nature of the study itself, with a change in the immunosuppression and overall management of kidney transplant recipients during time

Abstract

  • Please update accordingly once the manuscript is revised. 

Thank you

Reviewer 2 Report

Thank you for giving me an opportunity to review your manuscript titled “Living donor kidney transplantation improves graft and recipient survival in patients with multiple kidney transplants”. The concept of this manuscript has a potential which would lead the future perspective in the era of increasing in the number of multiple kidney transplantation. However, some critical problems are here which should be drastically amended. The reasons are below;

[major]

  1. This manuscript stated “living donor kidney has the greatest impact on improving graft and recipient survival in patients with multiple kidney transplants” in conclusion. This seems to be common in either single or multiple kidney transplantation. Furthermore, Figure 2b shows a significant difference between 1st, 2nd, and 3rd transplant by univariable analysis as KM (you concluded from here), however, multivariable analysis shows that not only living donor but also transplant numbers are significantly associated with death censored graft failure. This means no association between transplant number and living donor. Moreover, Table 3 shows graft survival in only 3rd+ transplants. Superior result in living donor is not surprising since living donor has low sensitization, high quality, shorter ICT, and etc…. If authors analyzed using sensitization variables as adjusted confounders, HLA mismatches, duration of dialysis before transplant, and time to transplant from 1st to 3rd, significant association of living donor would be decreased. The authors should state why did these confounders select in this analysis.

  1. Regarding recipient survival, 2nd and 3rd transplant recipients might be in the immortal bias. Since 2nd and 3rd recipients are more likely to be the stronger patients than died during their first transplant. The authors should state the possibility of immortal bias and do the sensitivity analysis using different points of starting measurement (such as all from 1st transplant).

  1. In general, low graft survival is related to worse patient survival. However, Figures 2 and 3 are not consistent with this theory. The authors should explain this discrepancy.

  1. Regarding allograft nephrectomy, previous reports revealed much higher rate of nephrectomy on re-transplantation. For instance, approximately 40% in Turkey (Transplant Proc. 2005 Sep;37(7):2957-61.) and 75% in the UK (NDT. 2009 Feb;24(2):639-42.). Why does this cohort contain only 4/333 (1.2%)? This discrepancy should be discussed in the discussion section. Nephrectomy might improve graft survival according to the meta-analysis (Nephrol Dial Transplant (2018) 33: 700–708). Ideally, nephrectomy should be added as an adjusted confounder.

  1. I cannot understand the necessity of 2nd paragraph in the introduction section for the study purpose of this study.

[minor]

  1. cRF is not common in the daily clinical practice outside the UK. Please explain the detailed or put the reference.

  1. Why was an earlier decade better for graft and patient survival?

  1. please state abbreviations for ABDR, MM in Table 2.

  1. Please describe which is 1st, 2nd, and 3rd transplant in Figure 3 and 2.

  1. Please define detailed RRT duration. Before 1st transplant? or total duration?

  1. Please add the first digit in HLA mismatch in 2nd (2+-1.4) and ICT in 3rd+ (13+-9.9) in Table 1.

Author Response

Thank you for giving me an opportunity to review your manuscript titled “Living donor kidney transplantation improves graft and recipient survival in patients with multiple kidney transplants”. The concept of this manuscript has a potential which would lead the future perspective in the era of increasing in the number of multiple kidney transplantation. However, some critical problems are here which should be drastically amended. The reasons are below;

We thank the reviewer for the time to review our manuscript and the valuable comments that significantly improve the quality of the work.

[major]

  1. This manuscript stated “living donor kidney has the greatest impact on improving graft and recipient survival in patients with multiple kidney transplants” in conclusion. This seems to be common in either single or multiple kidney transplantation. Furthermore, Figure 2b shows a significant difference between 1st, 2nd, and 3rd transplant by univariable analysis as KM (you concluded from here), however, multivariable analysis shows that not only living donor but also transplant numbers are significantly associated with death censored graft failure. This means no association between transplant number and living donor. Moreover, Table 3 shows graft survival in only 3rd+ transplants. Superior result in living donor is not surprising since living donor has low sensitization, high quality, shorter ICT, and etc…. If authors analyzed using sensitization variables as adjusted confounders, HLA mismatches, duration of dialysis before transplant, and time to transplant from 1st to 3rd, significant association of living donor would be decreased. The authors should state why did these confounders select in this analysis.

Thank you for your valuable comment. From the study of Dabare et al. (Outcomes in Third and Fourth Kidney Transplants Based on the Type of Donor. Transplantation, 2019. 103(7): p. 1494-1503) the long-term outcome of repeated kidney transplantation seemed to be independent from the donor type, but only determined by the recipient status, this is why we instead emphasised the importance of a high quality graft. Furthermore, the variables used in the multivariate analysis included HLA mismatches and duration of renal replacement therapy prior to transplantation. The association between a living donor transplant and graft and recipient survival remained statistically significant after these adjustments.

Regarding recipient survival, 2nd and 3rd transplant recipients might be in the immortal bias. Since 2nd and 3rd recipients are more likely to be the stronger patients than died during their first transplant. The authors should state the possibility of immortal bias and do the sensitivity analysis using different points of starting measurement (such as all from 1st transplant).

Thank you we have stated the possibility of immortal bias, due to the fact these patients are stronger.

  1. In general, low graft survival is related to worse patient survival. However, Figures 2 and 3 are not consistent with this theory. The authors should explain this discrepancy.

Thank you, we have stated that aside from younger and fitter recipients, we have selected better graft that lead to primary graft function with a significant improvement in the short- and long-term outcome

  1. Regarding allograft nephrectomy, previous reports revealed much higher rate of nephrectomy on re-transplantation. For instance, approximately 40% in Turkey (Transplant Proc. 2005 Sep;37(7):2957-61.) and 75% in the UK (NDT. 2009 Feb;24(2):639-42.). Why does this cohort contain only 4/333 (1.2%)? This discrepancy should be discussed in the discussion section. Nephrectomy might improve graft survival according to the meta-analysis (Nephrol Dial Transplant (2018) 33: 700–708). Ideally, nephrectomy should be added as an adjusted confounder.

Thank you, we have added the following: This caution with regard to nephrectomy is partly due to the immunological effect of graft nephrectomy, where nephrectomy and the cessation of immunosuppression can precipitate the development of HLA antibodies which limits access to retransplantation. In this series only four graft nephrectomies were recorded: one because of antibody mediated rejection with systemic involvement, one because of a bleeding catastrophe after the transplant and one to create space for a potential 4th kidney transplant. The limited number of graft nephrectomy at our institution is in contrast with other centres’ experiences, estimated approximately 40% from a Turkish report and up to 75% in a UK single centre experience from 2009 [30]. We tend to avoid as a general principle an additional operation, unless not strictly required, also because there is uncertainty whether it could influence the immunological status of the potential candidates and the likelihood of finding a suitable match, with antibody absorption from the graft itself. [28, 30]. As previously stated in fact, the graft nephrectomy would imply the cessation of the immunosuppression, giving rise to antibody production due to the persistence of donor antigen-presenting cells after the transplantectomy. Furthermore, with the evidence that HLA matching plays a fundamental role in the context of repeated kidney retransplantation, from the present study and other large registry analysis, we think that a synergistic approach to optimise the recipient condition and general immunological status would better satisfy an increased complexity at the moment of transplantation in the eventuality that a prolonged surgical time would be required to find a suitable implantation site. Once again, we emphasize the importance of a high-quality graft, like the one retrieved from a living donor.

  1. I cannot understand the necessity of 2nd paragraph in the introduction section for the study purpose of this study.

Thank you. We have expanded the introduction to clarify this point with regard to the challenges of the repeated kidney transplantation that might impact on the short and long-term outcomes, independently from the graft type.

[minor]

  1. cRF is not common in the daily clinical practice outside the UK. Please explain the detailed or put the reference.

Thank you, we have also mentioned the panel reactive antibody screening (PRA), more commonly used outside the UK. cRF and cPRA are equivalent.

  1. Why was an earlier decade better for graft and patient survival?

Thank you, this might be a sporadic finding with no statistical significance

  1. please state abbreviations for ABDR, MM in Table 2.

 Thank you we modified as following: HLA mismatch at HLA-A, B and DR.

  1. Please describe which is 1st, 2nd, and 3rd transplant in Figure 3 and 2.

 Thank you we added the following: Blu line: recipients of 1st kidney transplant; green line: recipients of 2nd kidney transplant; red line: recipients of third kidney transplant.

  1. Please define detailed RRT duration. Before 1st transplant? or total duration?

 Thank you, this is the total duration

  1. Please add the first digit in HLA mismatch in 2nd (2+-1.4) and ICT in 3rd+ (13+-9.9) in Table 1.

 Thank you

Round 2

Reviewer 1 Report

Thank you for providing revisions to the manuscript.

Author Response

Thank you for your review.

Reviewer 2 Report

Please define the below;

duration of RRT is sum of the RRT days or something. I cannot find the amended definition. Other suggestions are the same. 

The other would be fine for me. 

Author Response

Thank you for your suggestion.

We have added in the methods the following:

Duration of renal replacement therapy was expressed as cumulative time per month.